# Advanced Non-Clear Cell Renal Cell Carcinoma Treatments and Survival: A Real-World Single-Centre Experience

**DOI:** 10.3390/cancers15174353

**Published:** 2023-08-31

**Authors:** Davide Bimbatti, Francesco Pierantoni, Eleonora Lai, Melissa Ballestrin, Nicolò Cavasin, Elisa Erbetta, Chiara De Toni, Umberto Basso, Marco Maruzzo

**Affiliations:** 1Oncology 1 Unit, Istituto Oncologico Veneto, IOV-IRCCS, 35128 Padova, Italy; davide.bimbatti@iov.veneto.it (D.B.); eleonora.lai@iov.veneto.it (E.L.); nicolo.cavasin@iov.veneto.it (N.C.); elisa.erbetta@iov.veneto.it (E.E.); chiara.detoni@iov.veneto.it (C.D.T.); umberto.basso@iov.veneto.it (U.B.); 2Oncology 3 Unit, Istituto Oncologico Veneto, IOV-IRCCS, 35128 Padova, Italy; francesco.pierantoni@iov.veneto.it (F.P.); melissa.ballestrin@iov.veneto.it (M.B.); 3Department of Surgery, Oncology and Gastroenterology, University of Padua, 35128 Padova, Italy

**Keywords:** renal cell carcinoma, non-clear cell RCC, papillary RCC, immunotherapy combinations

## Abstract

**Simple Summary:**

Non-clear cell renal cell carcinoma (nccRCC) represents about the 20% of all RCCs but recommendations on treatment lacks evidence since the clinical trials include only clear cell RCC (ccRCC). The aim of our retrospective studies was to evaluate the efficacy of TKI and immunotherapy-based combinations in this population. We confirmed that nccRCC are heterogeneous and have a poorer prognosis as compared to ccRCC. The introduction of immunotherapy increased the efficacy of the treatments and the survival outcomes. Prognostic factors such as IMDC score or NLR are valid also for nccRCC. We highlight the importance of a pathological review and the need for prospective randomized trials designed for the different subtypes.

**Abstract:**

Background: Non-clear cell renal cell carcinoma (nccRCC) is a heterogeneous group of cancer. Treatment recommendations are extrapolated from ccRCC and lack solid evidence. Here, we review advanced nccRCC patients treated at our institute. Patients and methods: We collected retrospective data on all advanced nccRCC pts treated at the Istituto Oncologico Veneto from January 2008. We compared overall response rate (ORR), progression free survival (PFS) and overall survival (OS) according to histological subtypes and type of systemic treatments. Kaplan-Meier method, log-rank test and Cox regression were used to estimate and compare PFS and OS. Results: Of 1370 RCC patients, 289 had a diagnosis of nccRCC and 121 were eligible for the analysis. Fifty-three pts showed papillary histology (pRCC), 15 chromophobe; 37 unclassified RCC (NOS-RCC), 16 other histologies. Pts with chromophobe and other hystologies showed poorer survival rates compared to pRCC and NOS-RCC (mOS 10.7 vs. 20.7 vs. 30.7, *p* = 0.34). Pts treated with combination regimens achieved a better OS (30.7 vs. 13.7, *p* = 0.10), PFS (12.7 vs. 6.4, *p* = 0.10) and ORR (42.4% vs. 13.9%, *p* = 0.002) than those treated with monotherapy. IMDC and Meet-URO score retained their prognostic value. Conclusion: Our retrospective real-life cohort of advanced nccRCC patients shows that immunotherapy-based combinations could improve ORR, PFS and OS compared to TKI monotherapy. Prospective trials for nccRCC patients utilizing novel therapies are ongoing and their results eagerly awaited.

## 1. Introduction

Renal cell carcinoma (RCC) is the most common type of kidney cancer in adults. Although RCCs are more frequently detected in the earlier stages by incidental abdominal imaging, there is still a remarkable proportion of patients with synchronous metastases and another 30–40% of patients that will develop distant metastasis after surgery [1,2].

Historically, RCC was clinically treated as two distinct entities: clear cell renal carcinoma (ccRCC) and non-clear cell renal cell carcinoma (nccRCC). According to 2016 WHO classification, nccRCC represents a highly heterogeneous group which is about 20% of RCC [3,4]. More than 15 nccRCC entities are listed including papillary renal cell carcinoma (pRCC), fumarate-hydratase deficient renal cell carcinoma (fhRCC), chromophobe renal cell carcinoma (chRCC), collecting duct carcinoma (CDC), renal medullary carcinoma (RMC), MiT family translocation renal cell carcinomas (tRCC) and oncocytoma among others. pRCC is the most frequent subtype (10–15%), followed by chRCC (5%), CDC (1%), RMC (1%), and tRCC (1%) [3].

Papillary RCCs were usually subclassified into type 1 and type 2 tumors [5], but due to diagnostic inconsistency and clinical irrelevance, this subdivision is not recommended in the 2022 WHO [6]. Molecular alterations to mesenchymal-epithelial transition (MET) including mutations and amplification of the chromosome 7 are found in 15–20% of sporadic cancers [7] in addition to rare hereditary forms characterized by MET germline mutation [8]. MET plays a significant role in the pathophysiology of pRCC and could be a therapeutic target now being investigated in clinical trials [9,10].

chRCC, like the oncocytoma, are generally characterized by an excellent prognosis but in the rare metastatic cases the outcome is very poor. It usually harbor mutations in PTEN and TP53, whole chromosome loss, and TERT gene rearrangement [11,12].

CDCs are aggressive tumors usually presenting with synchronous metastases and a very poor prognosis [13]. Cytogenetic alterations of chromosomes 1q, 8p, and 13q can be found [14].

tRCC encompasses a spectrum of fusions involving TFE3 (on Xp12), TFEB, and MITB. It has a higher incidence in pediatric patients and most of these translocations result in aggressive biology and a poor prognosis [15].

Finally, RCC with sarcomatoid differentiation (sRCC) is not a distinct morphogenetic subtype of RCC. The presence of a sarcomatoid component can be observed with any RCC histology and usually indicates a more aggressive biological behavior [16].

Since pivotal trials included only patients with ccRCC, treatments for nccRCC are mainly guided from data extrapolated from ccRCC trials or from promising case reports and retrospective series.

Data on the efficacy of single-agent target therapy in nccRCC demonstrated moderate activity in terms of response rates and limited overall survival [17]; however, NCCN and ESMO guidelines still recommend these treatments as the first-line option together with enrollment in clinical trials [18].

Nowadays, combination therapies based on immune checkpoint inhibitors have markedly improved the treatment of ccRCC but data on nccRCC are still scarce, taken from small or retrospective trials [19,20] where this subgroup is often described as a single entity, while it is actually composed of numerous subtypes with distinct carcinogenesis, prognosis and probably treatment sensitivity.

The aim of this study was to review the historical series of patients treated at our institute to provide real-world data on the prognosis and efficacy of the different therapies used for the various histotypes of nccRCC. We also tried to put these data into perspective and compare them with those found in literature in order to reach some final considerations on the best possible management and therapy.

## 2. Materials and Methods

Patient data were examined to evaluate the real-world setting of advanced nccRCC Patients were defined advanced if they have a metastatic disease or a locally advanced disease not amenable to curative surgery or radiotherapy. All consecutive patients with nccRCC treated at the Istituto Oncologico Veneto between January 2008 and October 2022 were assessed retrospectively. The inclusion criteria included a histological diagnosis of nccRCC according to WHO 2016 and the availability of at least baseline clinical and demographic information. Patients with ccRCC with sarcomatoid differentiation were excluded. But patients with exclusive presence of sarcomatoid component in the absence of clear cell areas were considered. Clinical data were extracted from electronic patient records (EPRs), available for all the patients at the Istituto Oncologico Veneto. EPRs were used to ascertain demographic information, histology information (staging according to TNM, grading), the risk group according to the IMDC classification [21], duration of treatment, best response, progression free survival (PFS) and overall survival (OS). Tumor assessment was scheduled according to standard clinical practice, and response was measured using Response Evaluation Criteria in Solid Tumors (RECIST) version 1.1 [22]. For all the assessments, patients were divided into 3 groups according to histotype: pRCC, undifferentiated and not otherwise specified RCC (NOS-RCC) and others (all the other patients); and 2 groups according to the type of treatment received: combination treatment (treatment combinations based on immunotherapy) or monotherapy (TKI alone or chemotherapy). Despite the different mechanism of action we decided to group TKI alone and chemotherapy considering these as the old standard treatment to be compared with the new immune-based combination therapies. In addition only few patients had received chemotherapy as first line treatment.

The Meet-URO score was also assessed in our patient cohort: this is a 5-class score defined by a multivariable model incorporating Neutrophil-to-Lymphocytes Ratio (NLR), IMDC score and bone metastases [23].

Key metrics were summarized by means of descriptive statistics. PFS and OS were estimated using the Kaplan-Meier method. The log-rank test was used to compare patients’ PFS and OS in accordance with histotype, the IMDC risk group and Meet-URO score. The Chi-squared test was used to determine the association between each pair of variables. Results were deemed as statistically significant if their *p*-values were <0.05. All statistical analyses were carried out using “R” v.4.2.3

## 3. Results

### 3.1. Patients’ Characteristics

Of 1370 patients with RCC, 289 had nccRCC (21%). Of these patients, 121 were treated for advanced nccRCC in our single-centre setting between January 2008 and October 2022. The majority were male (65.3%), with a median age of 64 years old. The vast majority of patients were in good general condition (ECOG 0–1: 77.7%) at presentation. The patients’ general characteristics are summarised in Table 1.

The most prevalent histology was pRCC (*n* = 53, 43.8%). The second most common histological type was chRCC (*n* = 15, 12.4%), followed by a small number of CDC (*n* = 5, 4.1%) and other, rarer histologies such as RMC (*n* = 1, 0.8%), tRCC (*n* = 1, 0.8%), malignant epithelioid angiomyolipoma (AML) (*n* = 2, 1.6%) and pure sRCC (*n* = 7, 5.8%). Moreover, NOS-RCC was reported for 37 patients (30.6%).

Regarding the type of treatment received, of 121 patients, 13 were only deemed suitable for best supportive care (10.7%) due to an inadequate performance status. Three patients (2.4%) with a limited metastatic spreading of the disease were treated with surgery or radiotherapy of the metastatic sites. The remaining 105 (86.8%) patients were treated with first line systemic therapy. Three patients (2.9%) received platinum salts-based chemotherapy while 69 (65.7%) patients receive TKI monotherapy and 33 (31.4%) patients received (IO-IO and IO-TKI in 12 and 21 patients respectively). First-line therapies according to the histotype are reported in Table 2.

Eighty-eight patients progressed to first-line therapy during the follow-up period. Twenty-eight of these patients (31.8%) did not receive a second line treatment and were considered eligible only for best supportive care, in particular 11 out of 38 pRCC patients, 8 out of 24 NOS-RCC and 9 out of 26 patients with other histotypes. On the other hand, 20 patients (22.7%) received nivolumab in second line treatment while 2 patients (2.3%) received chemotherapy and the other 38 (43.2%) a TKI (mainly Cabozantinib and Sunitinib). Furthermore, at the time of this analysis, 22 (20.9%) received three or more lines of systemic therapy.

### 3.2. Treatment Outcomes

Median follow up of the entire population was 47.3 months (95% CI, 34.7–96.8). In particular, median follow up was 78.5 months (95% CI, 39.0–NA) for patients that received monotherapy treatment and 21.9 months (95% CI, 19.1–NA) for them that received immune-based combination treatment.

Best response to first-line therapy was evaluable in 103 patients. Of these 103 patients, 38 (36.9%) showed progressive disease at the first radiological evaluation. On the other hand, the best response was stable disease in 41 (39.8%) and partial response in 21 (20.4%) patients. Three (2.9%) complete responses were detected in our single-centre setting. One with pRCC, one with NOS-RCC and one with sRCC; two of them received a combination treatment but all were characterized by a single site (lung or lymph nodes) disease. ORR was superior for combination treatment than monotherapy in the entire population (42.4% vs. 13.9%, *p* = 0.002).

The first-line median PFS was 7.7 months (95% CI, 6.0–11.5), while the median OS was 18.5 months (95% CI, 13.3–33.2) for the 105 patients treated with systemic therapy. Comparing tumor histology, pRCC had an mPFS at the first-line treatment of 9.7 months (95% CI, 6.1–15.1), NOS-RCC had a mPFS of 9.2 (95% CI, 6.5–18.9) while other RCCs had a mPFS of 3.0 months (95% CI, 2.4–12.7) (*p* = 0.03). However, the mOS did not significantly differ between the three groups: mOS 20.7 months (95% CI, 13.7–42.4) vs. 30.7 (95% CI, 10.9–NR) vs. 10.7 (95% CI, 4.0–38.0) respectively (*p* = 0.34) (Figure 1).

On the other hand, comparing the type of treatment received, patients treated with first-line immune-based combination therapy compared to those treated with first-line monotherapy had a slightly better mPFS (12.7 months, 95% CI: 9.2–27.7 vs. 6.4 months, 95% CI: 3.8–10.0, *p* = 0.10) and mOS (30.7 months, 95% CI: 8–33 vs. 13.7 months, 95% CI: 9.8–28.9, *p* = 0.10) (Figure 2).

A total of 53 patients received first- or second-line immunotherapy. The mOS of patients receiving immunotherapy during their clinical history was significantly higher than those who had never received it: 32.4 months (95% CI, 19.2–48.5) vs. 9.8 months (95% CI, 7.2–21.7), *p* = 0.01 (Figure 3). Results of ORR, mPFS and mOS according to histotype and type of treatment are reported in Table 3.

In our cohort, 79 patients were metastatic at diagnosis but only 67 of these patients were treated. The PFS and OS were better for the 35 patients who underwent to a cytoreductive nephrectomy before the treatment compared to the 32 patients who received systemic treatment immediately (mPFS 7.2 months, 98% CI: 4.5–26.0 vs. 3.0 months, 95% CI: 2.3–8.5, respectively, *p* < 0.01; mOS 19.2 months, 95% CI: 13.3–44.1 vs. 8.4 months, 95% CI: 4.0–32.4, respectively, *p* = 0.05). In comparison to these results, the 38 remaining patients with metachronous disease who received systemic treatment at least 6 months after the nephrectomy (1 year for the majority) had a significantly longer PFS and OS (mPFS 12.0 months, 95% CI: 8.0–18.3; *p* = 0.002 and mOS 38.1 months, 95% CI: 20.7–NA; *p* = 0.003).

### 3.3. Prognostic Factors

We evaluated prognostic factors such as International Metastatic RCC Database Consortium (IMDC) score, neutrophil-to-lymphocyte ratio (NLR) and Meet-URO score as well as the number of metastatic sites. The IMDC score confirmed its prognostic role even in this setting: the mOS for patients classified as having a good, intermediate or poor risk was 48.5, 32.4 and 3.8 months, respectively (*p* < 0.01) while the mPFS was 13.5, 8.6 and 2.5 months respectively (*p* < 0.01).

The NLR was evaluable for 77 treated patients; using a cut-off of 3, it demonstrated significant differences between the two groups: patients with a NLR < 3 had a significantly better PFS and OS than patients with an NLR ≥ 3 (mOS 20.7 months, 95% CI: 12.6–40.5 vs. 5.4 months, 95% CI: 3.9–22.7, respectively, *p* < 0.01; mPFS 9.7 months, 95% CI: 6.5–15.4 vs. 3.8 months, 95% CI: 2.7–11.4, respectively, *p* = 0.03).

The Meet-URO score was available for all the patients. Ten out of 33 patients (30.3%) in class 4 and 5 did not received any treatment. These data confirm the poor prognosis of these patients and, in part due to the low number of treated patients in these classes, allow us to combine class 4 and 5 for the survival analysis. So, taking into consideration 4 categories of the Meet-URO score the mPFS was 11.5, 12, 6.8 and 2.4 months respectively (*p* < 0.01) and the mOS was 63.7, 33.2, 23.1 and 3.7 months respectively (*p* < 0.01).

Finally, we assessed the prognostic value of the number of metastatic sites. We found that there was a significant trend towards decreased OS rising from 1 to 2 to ≥3 sites involved (mOS 23.6 vs. 15.5 vs. 11.6 months respectively, *p* = 0.07).

## 4. Discussion

Despite examining all the cases of advanced/metastatic nccRCC at a high-volume institute, patient numbers were relatively low. Probably, a smaller proportion of nccRCC patients than those with ccRCC will develop metastatic disease and will be referred to an Oncology Department after complete surgery of the primary cancer. In fact, it is well-known that the most common non-clear histotype, pRCC and chRCC, usually has a better prognosis and a lower risk of cancer-associated death than ccRCC when diagnosed in a localized stage [12,24]. Furthermore, given that no adjuvant therapies are available for this subgroup of patients, the follow up could be performed without an oncological referral.

However, when locally advanced or metastatic, nccRCC seems to have a worse prognosis, compared to ccRCC [24]. In our case series, in fact, mPFS and mOS are about 8 and 18 months respectively without considering patients untreated because suitable only for best paliative care. In addition, mPFS and mOS were even shorter for patients with histotypes other than pRCC or NOS-RCC. This difference highlights the importance of the new WHO classification where, for example, the division between type 1 and 2 in pRCC is no longer recommended and the Fumarate Hydratase deficient RCC is now considered a unique entity [6]. Furthermore, considering all the prognostic and therapeutical implications, this emphasizes the importance of an expert pathological review. In our retrospective cohort, the incidence of different nccRCC histologic subtypes was comparable to published data, with a higher ratio of NOS-RCC because many cases were not reviewed [25]. This group of patients had a mPFS and mOS similar to the pRCC (where type 1 and 2 were included indistinctly) but it is probably composed of many different subtypes with likely very different sensitivities to the treatments and certainly very different outcomes.

The majority of patients (65.3%) in our single-centre study were metastatic at diagnosis and about half of them (46.8%) underwent cytoreductive nephrectomy. The efficacy of this conventional practice has been questioned by the results of the CARMENA trial [26]; however, it was standard practice in the period in which our patients were diagnosed. It should be noted that only ccRCC patients were included in the CARMENA trial, but some retrospective evidence, in line with our results, suggests a role for cytoreductive surgery in selected patients with metastatic nccRCC [27].

As expected, most of the patients who received systemic first-line therapy were treated with only a tyrosine kinase inhibitor (TKI), predominantly Sunitinib. PFS and OS in these patients were similar to those reported in literature [9,10,28,29,30,31,32,33,34]. Significant TKI data are being accumulated in these trials with nccRCC patients: in the ESPN trial, Sunitinib performed slightly better (median OS 16.2 vs. 13 months) [30], while Cabozantinib showed promising results in the SWOG 1500 trial [10]. In our historical cohort, however, only 13 patients received Cabozantinib and the group size is too small for any statistical inferences. Finally, the MET inhibitors Savolitinib and Foretinib demonstrated promising data in nccRCC in phase II trials [10,35,36]. However, despite the fact that the phase III trial SAVOIR confirmed the encouraging efficacy of Savolitinib, small patient numbers and short follow-up were unsuitable to reaching robust conclusions [33]. In summary, as indicated by the literature [37], the response rate is poor and the majority of the patients treated with a TKI only progressed at their first or second assessment.

Therefore, there has been an increase in data on immunotherapy and immuno-based combination therapy in recent years [20]. When used as single agents, Nivolumab [38] and Pembrolizumab [39] demonstrated short PFS (2.2–4.2 months) but promising ORR (13.6 and 26.7% respectively). Due to these data, IO-based combinations like Ipilimumab-Nivolumab [40], Atezolizumab-Cabozantinib [41], Nivolumab-Cabozantinib [42], Durvalumab-Savolitinib [43] and Pembrolizumab-Lenvatinib [44] were tested in prospective trials. The results of these trails are very promising, with PFS varying from 3.7 to 12.5 months and ORR ranging froffm 19.6% to 49%.

In our findings, patients who were treated with a first-line immunotherapy combination had a PFS of 12.7 months and an ORR of 42.4%. These results are in line with those reported in the prospective trials described above but, more importantly, are similar to those recently reported in two real-world experiences [25,45]. Furthermore, we can also report that patients who received immunotherapy in their history achieved longer OS than those who did not. This result confirms that immunotherapy could be an effective option for nccRCC patients and combination treatment in first line is a promising strategy in this subgroup of patients. Nevertheless, there are different outcomes based on the histologic subtypes, which require further investigation. It is important, in fact, that future randomized trials take into consideration the differences between the various histotypes, limiting the rate of NOS-RCC, in order to understand the individual sensitivities to treatment and implement biology-tailored management.

Fortunately, prognostic scores appear to maintain their ability to stratify patients even in nccRCC. In our cohort, we found that the IMDC score confirmed its strong prognostic role as expected. In fact, between 5% and 13% of nccRCC patients were included in both the first IMDC score report [21] and in the external validation group [46]. Notably, non-clear cell histology was associated with poor OS. Nevertheless, the score is still significant in nccRCC patients, comparable to data previously published by Kroeger et al. [47]. We also tested in our cohort the Meet-URO score, which was implemented in collaboration with our research group [23] and validated in patients treated with either Nivolumab or Cabozantinib in second-line treatment [48]. Even in an unselected cohort of nccRCC patients, the Meet-URO score retained its prognostic value and was able to stratify patients accurately according to their prognosis.

Finally, the NLR score also proved valuable in our cohort of nccRCC patients. The negative prognostic value of the NLR, in fact, is largely known in ccRCC but only few trials include and consider nccRCC patients [49,50]. These trials, in line with our report, verify that the NLR retains its value in this population but also confirm that patients with nccRCC have a higher NLR compared to patients with ccRCC [49].

Our trial has several limitations, including the retrospective collection of data with the possible risk of selection bias, the short follow-up, especially for the combination treatments, the small number of patients treated and the lack of a central pathological review. Nevertheless, this trial, like that reported by Izarn et al. [25], reflects what really happens in a referral center and all the challenges that an oncologist has to face when dealing with an nccRCC patient. These limitations, in fact, underline the significance of the pathological classification and the need to perform an expert review in order to reach a correct diagnosis and take an appropriate treatment decision. In addition, this work emphasizes the importance of immunotherapy and, overall, the importance of prospective randomized clinical trials, which confirm these results, to improving the standard of care and the survival outcomes in this heterogeneous and poor prognosis population.

## 5. Conclusions

NccRCCs are uncommon but account for about 20% of all RCCs. Our real-world data are comparable to those published in the literature for nccRCC and demonstrated that immunotherapy has led to an improvement in the prognosis, especially when used in first-line combination treatments.

Nonetheless, advancements in histopathology and molecular genetics are necessary in order to open new horizons for therapeutic options; while the results of the ongoing prospective selective trials for nccRCC utilizing IO and new targeted agents are urgently required to further improve patient outcome. In future, we recommend studies tailored to individual histologies because the nccRCC is group is too heterogeneous and, to date, this “umbrella” definition is limiting and should already be viewed as outdated.

## Figures and Tables

**Figure 1 cancers-15-04353-f001:**
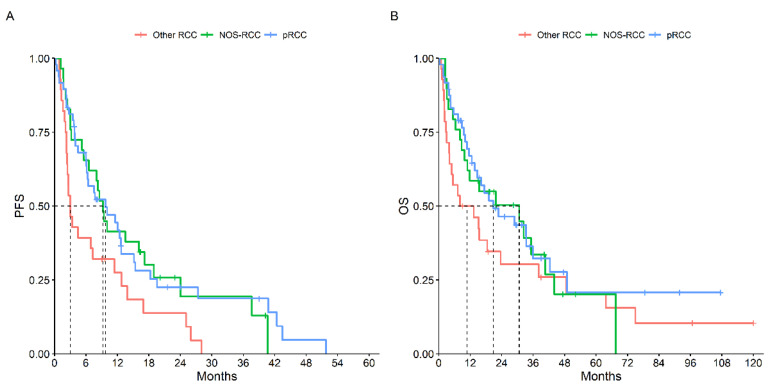
PFS (**A**) and OS (**B**) According to Histotype.

**Figure 2 cancers-15-04353-f002:**
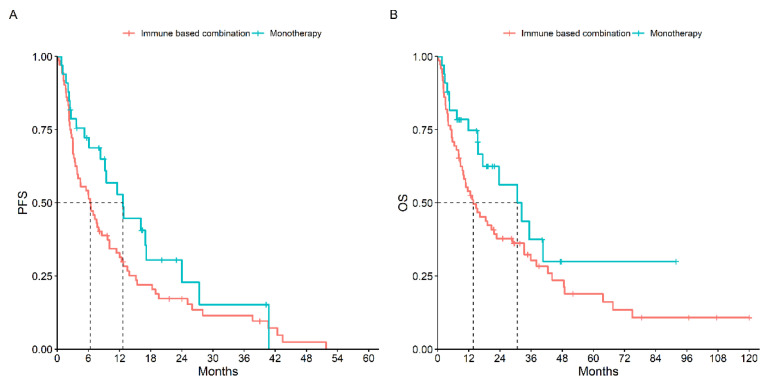
PFS (**A**) and OS (**B**) According to Type of Treatment Received.

**Figure 3 cancers-15-04353-f003:**
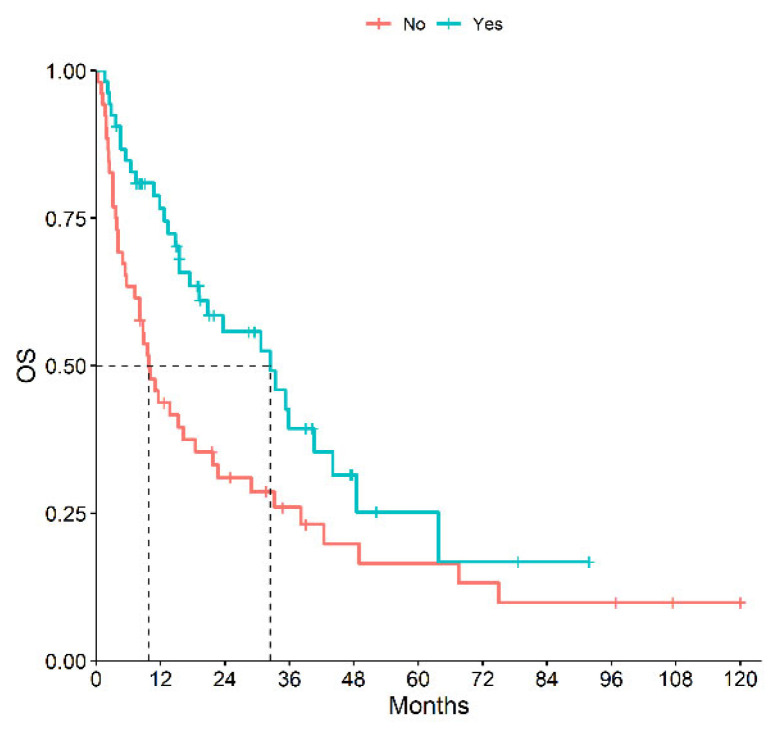
OS of Patients Receiving or Not Immunotherapy during Their History.

**Table 1 cancers-15-04353-t001:** The patients’ general characteristics, including the IMDC classification and ECOG PS.

Characteristics	N (%)
Age (median, IQR)	64 years (57–74)
Female/Male	42 (34.7%)/79 (65.3%)
*Nephrectomy*	
Metachronous	42 (34.7%)
Cytoreductive	37 (30.6%)
*IMDC*	
Good Risk	29 (24.0%)
Intermediate Risk	61 (50.4%)
Poor Risk	31 (25.6%)
*PS ECOG*	
0	68 (56.2%)
1	26 (21.5%)
≥2	27 (22.3%)
Time From Diagnosis to Metastasis <12 Months	87 (71.9%)
*Number of Metastatic Sites*	
1	46 (38.0%)
2	46 (38.0%)
>2	29 (24.0%)
*Site of Metastases*	
Lung	61 (50.4%)
Lymph Node	54 (44.6%)
Liver	35 (28.9%)
Bone	29 (24.0%)
Soft Tissue	13 (10.7%)
Adrenal Gland	11 (9.1%)
Brain	5 (4.1%)
Pancreas	4 (3.3%)
Other	10 (8.3%)
*NLR*	
*<3*	48 (39.7%)
*≥3*	38 (31.4%)
*NA*	35 (28.9%)
*Meet Uro Score*	
1	22 (18.2%)
2	43 (35.5%)
3	23 (19.0%)
4	23 (19.0%)
5	10 (8.3%)

*N* = number of patients.

**Table 2 cancers-15-04353-t002:** First-line Systemic Therapies.

Histotype (N = 105)	TKI Monotherapy (*n* = 69)	Combination (*n* = 33)	Chemotherapy (*n* = 3)
pRCC (*n* = 48)	Cabozantinib (*n* = 6)	Axi-Pembro (*n* = 6)	Carboplatin-Nab-Paclitaxel (*n* = 1)
	Pazopanib (*n* = 9)	Cabo-Nivo (*n* = 1)	
	Sunitinib (*n* = 19)	Savo-Durva (*n* = 1)	
	Tivozanib (*n* = 1)	Ipi-Nivo (*n* = 4)	
Other RCC (*n* = 28)			
chRCC (*n* = 13)	Cabozantinib (*n* = 3)	Axi-Pembro (*n* = 3)	
	Pazopanib (*n* = 1)		
	Sorafenib (*n* = 1)		
	Sunitinib (*n* = 5)		
CDC (*n* = 5)	Cabozantinib (*n* = 1)		Platinum Based CT (*n* = 2)
	Sunitinib (*n* = 2)		
sRCC (*n* = 7)	Cabozantinib (*n* = 1)	Axi-Pembro (*n* = 2)	
	Pazopanib (*n* = 1)		
	Sunitinib (*n* = 3)		
RMC (*n* = 1)	Sunitinib (*n* = 1)		
AML (*n* = 1)	Everolimus (*n* = 1)		
tRCC (*n* = 1)		Lenva-Pembro (*n* = 1)	
NOS-RCC (*n* = 29)	Cabozantinib (*n* = 2)	Axi-Pembro (*n* = 7) Ipi-Nivo (*n* = 8)	
	Pazopanib (*n* = 7)		
	Sunitinib (*n* = 5)		

Axi-Pembro = Axitinib + Pembrolizumab; Ipi-Nivo = Ipilimumab + Nivolumab; Cabo-Nivo = Cabozantinib + Nivolumab; Savo-Durva = Savolitinib + Durvalumab; Lenva-Pembro = Lenvatinib + Pembrolizumab; CT = chemotherapy; N = number of patients.

**Table 3 cancers-15-04353-t003:** Treatment Outcomes According to Histotype and Treatment Type.

Histotype/Treatment Type	ORR (%)	mPFS (Months, 95% CI)	mOS (Months, 95% CI)
All/All (*n* = 105)	23.3	7.7 (6.0–11.5)	18.5 (13.3–33.2)
All/combo (*n* = 33) All/mono (*n* = 72)	42.4 13.9	12.7 (9.2–27.4) 6.4 (3.8–10.0)	30.7 (17.4–NA) 13.7 (9.8–28.9)
pRCC/All (*n* = 48)	20.8	9.7 (6.1–15.1)	20.7 (13.7–42.4)
pRCC/combo (*n* = 12) pRCC/mono (*n* = 36)	41.7 13.9	12.7 (6.1–NA) 7.6 (4.5–15.1)	NA (17.4–NA) 20.7 (12.6-42.4)
NOS-RCC/All (*n* = 29)	34.5	9.2 (6.5–18.9)	30.7 (10.9–NA)
NOS-RCC combo (*n* = 15) NOS-RCC mono (*n* = 14)	46.7 21.4	16.1 (8.3–NA) 7.2 (3.2–37.6)	32.4 (30.7–NA) 9.2 (6.4–NA)
Other/All (*n* = 28)	14.3	3.0 (2.4–12.7)	10.7 (4.0–38.1)
Other/combo (*n* = 6) Other/mono (*n* = 22)	33.3 9.2	7.7 (2.3–NA) 3.0 (2.3–11.4)	15.5 (2.8–NA) 7.6 (4.0–48.5)

Combo = immune-based combination treatment; Mono = monotherapy.

## Data Availability

The data presented in this study are available on request from the corresponding author. The data are not publicly available due to privacy and ethical reasons.

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
