# Peer review of "Advanced Non-Clear Cell Renal Cell Carcinoma Treatments and Survival: A Real-World Single-Centre Experience"

_cancers, 2023, doi:10.3390/cancers15174353_

Round 1

Reviewer 1 Report

Please find the attached document, thank you.  

Author Response

The authors conducted retrospective single center study focusing on the real-world survival outcomes in patients with advanced non-clear RCC. The manuscript is well-written and worth publishing owing to sparse evidence regarding the efficacy of systemic therapy for non-clear RCC. The reviewer raised some suggestions as below

Many thanks for the appreciation.

Here the answers to your comment:

・Line 97: Please define “advanced RCC”, such as “not amenable to curative surgery or radiation therapy or metastatic RCC

We added a more precise definition.

・Line 106: Please insert abbreviations “OS” and “PFS”

Done

・Line 121: Please spell out “NLR”

Done

・Table1: Please provide the information on metastatic site.

We added it in the table with a specific session. Thanks for this advice.

・Line 166: Please mention the detailed pathologic subtypes in patients who achieved CR

We have given an insight in the results.

・Line 171-173: How did authors calculate the p-value of three-arm comparison? please mention in the method in more detail. (as well as some analyses in the manuscript)

The method applied to calculate the p-value is the log-rank test and it is used to compare Kaplan-Meier curves. This method provides an overall p-value comparison. It was already mentioned in the text but with specify that the methods was used also for comparing histotype.

・Line 197: maybe better to put “ such as” after “prognostic factors”

We agreed and modified it

・Lines 215-216: This statement is not correctly reflected the results. I mean that mOS was decreased according to the numbers of metastases (1: 23.6, 2: 15.5, >3: 11.6 months, respectively), while it did not reach statistical significance. Please change the statement.

Of course you are right. We corrected it

・Line 298: Please clarify the follow-up duration of both treatment arms in the results.

We added mFU of the whole cohort and of the single arms in the results.

Reviewer 2 Report

The authors showed real-world data about the survival outcomes of systemic therapies for metastatic nccRCC. The present study should bring a significant insight into therapeutics for clinicians. However, there are some concerned to be addressed.

1.       The authors should simplify the introduction in the manuscript, because it is not a review of nccRCC.

2.       It is not easy to understand the following sentence;

“Patients with ccRCC with saromatoid differentiation were excluded if not exclusive” (page 3, line 101)

3.       Why was sRCC included in the present study?

4.       Please revise Table 1 and 2 to make them easier to see.

5.       What is “N°” in Table 1?

6.       Complete response is rare in systemic therapies for patients with metastatic nccRCC. Please describe the details in the manuscript.

7.       Although TKI monotherapy and chemotherapy are included as mono in Table 3, it is inappropriate. These mechanisms for anti-cancer are different.

8.       Please show survival curves using the Kaplan-Meier method. It is easy for readers to understand the results.

9.       It is not easy to understand the following sentence;

“Fortunately, prognostic scores are a staple also in nccRCC” (page 7, line 282)

10.   Please describe the follow-up period in the present study.

11.   Which does the authors want to make a point as a prognostic model, Meet-URO score or NLR? 

Please see my comments.

Author Response

To reviewer 2

The authors showed real-world data about the survival outcomes of systemic therapies for metastatic nccRCC. The present study should bring a significant insight into therapeutics for clinicians. However, there are some concerned to be addressed.

Many thanks for your precise revision and suggestions.

Here the answers to your comment:

  1. The authors should simplify the introduction in the manuscript, because it is not a review of nccRCC.

We revised and simplified it by cutting parts where we made the list of histotypes

  1. It is not easy to understand the following sentence.

“Patients with ccRCC with sarcomatoid differentiation were excluded if not exclusive” (page 3, line 101)

We clarified that in the text.

  1. Why was sRCC included in the present study?

See answer to comment 2.

  1. Please revise Table 1 and 2 to make them easier to see.

The graphical aspect of the tables was edited by the Cancers Editorial Office. Personally I think that is pretty clear both for the content and for the aspect. But if there is something specific please let us know so we can work on it

  1. What is “N°” in Table 1?

N stay for Number of patient. We added this explanation.

  1. Complete response is rare in systemic therapies for patients with metastatic nccRCC. Please describe the details in the manuscript.

We have given an insight in the results.

  1. Although TKI monotherapy and chemotherapy are included as mono in Table 3, it is inappropriate. These mechanisms for anti-cancer are different.

We considered TKI alone and chemo as the old standard to compare with immune-based combo (the new possible standard). In addition only 3 patients received chemotherapy. We add this consideration in material and methods.

  1. Please show survival curves using the Kaplan-Meier method. It is easy for readers to understand the results.

We added some significant KM curves

  1. It is not easy to understand the following sentence;

“Fortunately, prognostic scores are a staple also in nccRCC” (page 7, line 282)

We modified and clarified it

  1. Please describe the follow-up period in the present study.

We added mFU of the whole cohort and of the single arms in the results.

  1. Which does the authors want to make a point as a prognostic model, Meet-URO score or NLR?

In the discussion we just wanted to underline that IMDC, NLR and MEET URO are valid prognostic factors in nccRCC as previously reported only in few studies. Of course, there is no one model that is better or clinically useful than another and to evaluate this aspect is not an aim of our work.

Round 2

Reviewer 2 Report

There are no additional comments.